# Virosomes: Beyond Vaccines

**DOI:** 10.3390/life15101567

**Published:** 2025-10-08

**Authors:** Hadeel K. Salameh, Mohammed M. Safi, Rafik Karaman

**Affiliations:** 1Pharmaceutical Sciences Department, Faculty of Pharmacy, Al-Quds University, Jerusalem 20002, Palestine; hsalama@staff.alquds.edu (H.K.S.); abusafi1999@hotmail.com (M.M.S.); 2Department of Sciences, University of Basilicata, Via dell’Ateneo Lucano 10, 85100 Potenza, Italy

**Keywords:** virosome, patents, hemagglutinin, neuraminidase, IRIV, liposomes

## Abstract

**Background:** One of the primary strategies for preventing and reducing infectious diseases is vaccination. There are numerous licensed vaccinations of various kinds that can prevent viral infection by triggering the immune system’s reaction to specific antigens beforehand. To elicit a stronger immune response, however, two elements of the immune system—humoral and cellular immunity—should be addressed. Since they target proteins that are difficult to alter, recent innovative techniques for vaccine delivery systems—such as liposomes, nanogels, microemulsions, etc.—have shown excellent immunogenicity qualities. **Methods:** PubMed, ScienceDirect, and Google Scholar were used as the databases for literature search, and keywords such as “Virosomes”, “Hemagglutinin”, and “IRIV” were selected to ensure relevant articles were included. **Results:** This article examines a cutting-edge method called virosomes, which are an effective way to deliver pharmaceutically active ingredients that target a variety of illnesses and ailments, as well as vaccines. This resulted from the fact that virosomes possess numerous structural characteristics that might trigger sophisticated immune reactions by utilizing the inactivated virus’s envelope or by imitating it through recombinant methods. **Conclusions:** Here, we will walk you through the history of virosome development, explore various manufacturing techniques, provide an overview of the latest patents, and conclude with the potential for more virosomal revolutions.

## 1. Introduction

### 1.1. Development of Vaccines

A vaccination is a preventive measure that shields people against infectious diseases linked to high rates of morbidity and death, such as smallpox, which has been completely eradicated due to a successful vaccine formula [1]. Through the acquisition of the immune system of the killed or weakened pathogen in advance, vaccination seeks to protect the human body from a particular infection. By shielding vaccinated people from the initial virus, this preventive measure can minimize the frequency of illnesses and, consequently, the risk of the disease spreading from person to person. There are several methods for making vaccines, and they fall into the following categories: Pathogen components, including polysaccharides, conjugated polysaccharides, or virus-like particles; live attenuated pathogens; inactivated pathogens; toxoids (i.e., inactivated toxins produced by the disease-causing bacteria); and natural or recombinant proteins [2]. Conventional vaccinations, like the influenza virus, can be problematic since viruses can change and make the vaccine useless [3]. Traditional vaccines are rendered ineffective due to rapid alterations in the virus’s envelope and immunologic escape caused by mutations the virus performs on the ligand of antibodies (antibody-mediated) [4]. Activating cytotoxic T-lymphocytes (CTLs) is a key step in vaccination, and depending on the outcome, antigens must elicit a CTL response to maximize vaccination by eliciting immunogenicity through vaccines that can initiate both humoral and cellular immunity [5]. Targeting proteins that are essential to the virus’s operation and seldom mutate (such as functional surface proteins, DNA, RNA, carbohydrates, etc.) can help solve these issues [6,7,8]. The inability of these subunits to elicit the immune system on their own led to the need for a compatible delivery system that adjuvanted the subunit. Examples of such adjuvants include cytosine phosphoguanine (CpG) motifs (a synthetic form of DNA that mimics bacterial and viral genetic material), saponins, etc., which immune cells can recognize [8,9], and nano-vaccine delivery systems like liposomes, microemulsions, nano-gels, dendrimers, and virosomes [10]. Nano-vaccines of a size range less than 500 nm (mostly 40–50 nm) can successfully stimulate cellular immunity by provoking responses of T CD8+ CD4+ Type 1 cells [11]. This focused attention on researching and creating these new vaccination delivery methods. Stegmann et al. have proposed a therapeutic cancer vaccine using virosomes, where synthetic long peptides (SLP) were added to the envelope of virosomes to instigate tumor HPV16 E6/E7+ eradication. It was noted that the CD8+ T cell response in mice was enhanced [12].

### 1.2. Overview of Virosomes

It was necessary to construct nano-vaccines to boost a high immune response, as they represented a novel method for vaccine delivery systems. Reconstituted from a phospholipid bilayer combined with the envelope of the inactivated virus, virosomes are nonvaccine, unilamellar vesicles (60–200 nm) that lack a nucleocapsid. The adjuvanted membrane subunits of an original or recombinant virus on the liposomal layer are what distinguish virosomes from other viruses, as both arms of the immune system can be efficiently activated by these subunits [8,13,14]. The robust advantages of virosomes, such as industrial-scale up, compatibility with good manufacturing practices (GMP), and high immunogenicity, have captured researchers’ attention in the development of efficient vaccines against infection [15]. Furthermore, virosomes’ structure makes them an effective drug delivery mechanism for a variety of medications, including vaccines, anti-cancer medications, fungal infection treatments, and anti-Alzheimer’s medications [16,17,18]. Nonetheless, the FDA approved several virosomal vaccines due to their high immunogenicity and safety when compared to other vaccinations, including the hepatitis A vaccine Epaxal^®^ [19,20,21], the influenza vaccine Invivac^®^, and Inflexal^®^. While many additional vaccines are still in clinical trials that indicate strong promise for impending approval, such as Hepatitis C, Mumps, malaria, etc. [22,23,24,25].

This review provides a thorough analysis of the structure, benefits, patents, manufacturing methods, and uses of virosomes. This makes the review a comprehensive reference for readers about the virosomes’ journey, starting from how they were discovered and ending with how they emerged beyond viral vaccines. Additionally, the thorough study of virosomes’ structure, properties, and various successful applications has supported the development of prospects and recommendations for virosomes as a potential delivery system for numerous applications and treatments in the future.

## 2. Materials and Methods

Information from this review was explored using databases such as PubMed, ScienceDirect, and Google Scholar. Keywords related to the review topic, such as “Virosomes”, “Vaccines”, “IRIV”, “Hemagglutinin”, etc., filters such as clinical trials, systematic reviews, meta-analysis, and randomized controlled trials, were also applied to increase the precision of the search in the databases. Full articles and peer-reviewed articles were also included in the search. Moreover, most recent studies were targeted to make sure the review article is up to date. Any article with a missing full text and any article that does not relate directly to the topic of Virosomes was excluded.

## 3. Structure of Virosomes

### 3.1. The Basic Structure of Virosomes

With a diameter of around 150 nm, virosomes are unilamellar, spherical vesicles that lack genetic material and are instead filled with the material of choice. Viral envelope proteins, which offer virosomes an edge over liposomes and enable them to fuse with immune cells and start their response, are mostly found in their structure together with liposomal carriers. The liposome is originally a phospholipid bilayer; it consists mainly of phosphatidylcholine (lecithin) and phosphatidylethanolamine (cephalin), which form approximately 70% of the virosome’s structure [8]. While the rest of the virosome structure (30%) consists of functional glycoproteins, known as spike proteins [8], which are hemagglutinin (HA) and neuraminidase (NA), embedded in the liposomal phospholipid (PL) bilayer. However, the components of the phospholipid bilayer and glycoproteins can be adjusted according to the compound that will be incorporated into the virosomal delivery system. For instance, the hydrophilic agent will be inoculated inside the cavity, while a hydrophobic agent will be embedded in the liposomal membrane of the virosome [13,26]. Figure 1 represents the virosome structure.

### 3.2. Roles of Components in Virosomes Activity

As previously stated, the liposome serves as a carrier in the virosome’s structure. Since they can overcome the poor pharmacokinetic (PK) properties of pharmaceutical agents (PA), such as low solubility, suboptimal distribution to the targeted site, and rapid clearance of PA, liposomes are primarily used as a delivery system for PA. By encapsulating PA, these issues are resolved, leading to good overall PK properties [27,28]. However, liposomes have suboptimal fusion ability with cells, giving rise to a new idea of combining spike proteins with the liposome structures mentioned in the previous part. First, the interaction between HA and the antigen-presenting cells (APCs) is potentiated by multiple sialic acid residues residing on APCs that bind to the HA-binding site [29]. It is crucial to mention that HA is a trimer glycoprotein consisting of two glycopolypeptides, HA1 and HA2. In the beginning, HA is synthesized as a precursor HA0, and the host cell is responsible for its cleavage into HA1 and HA2 subunits by N-glycosylation. The amino-terminal of the HA1 subunit contains the receptor-binding site on its globular head, bonded to the carboxy-terminal fragment HA2, which is membrane-anchored and responsible for fusion [30]. Figure 2 illustrates HA cleavage. At pH 7, the two subunits are linked by a disulfide bond in a stable inactive form [8]. At pH 5 to 6.5, which is the pH of endosomes [31], HA subunits refold extensively and irreversibly by exposing the hydrophobic regions of HA2, bringing the virosome and cells close enough to facilitate their fusion [8,29]. Then, the virosomes release their content into the cytoplasm of the cell [31]. Second, the other glycoprotein that renders virosomes a unique delivery system is NA, a mushroom-shaped tetramer of identical subunits, that works by removing sialic acid from cellular and virosomal glycoprotein (HA) [32]. This happens by catalyzing the cleavage of N-acetylneuraminic acid (sialic acid) from bound sugar residues, resulting in a decreased viscosity of the host’s mucus and allowing the virosome’s easier access to epithelial cells [20]. As a result, it gives the space for virosomes to spread to new cells.

## 4. Interaction of Virosomes with the Immune System

### 4.1. Mechanism of Action

The fusion capabilities of virosomes indicate that they can be encased within APCs and that an acidic pH promotes virosome-endosome fusion, which allows them to enter the lysosomal/endosomal pathway. To activate CD4+ helper T-cells, they are subsequently converted into peptides and coupled with the major histocompatibility complex (MHC) class II that is displayed on the cell surface. In addition, endogenous antigens synthesized within endosomes in the cytosol of APCs are processed by the proteasome, then transferred by transporters of antigenic peptides (TAP) into the endoplasmic reticulum to associate with MHC class I molecules. Lastly, this complex is presented at the surface of APC so it can stimulate CD8+ helper T-lymphocytes, which mature and proliferate into CTLs [8,33]. On the other hand, virosomes can be recognized by membrane-associated immunoglobulin (Ig) receptor molecules on B lymphocytes, then rearrange to interact with them causing strong activation of B-cells, this happens by processing virosomes into antigens, so they can then be presented to T cells by MHC II complex, this activates T cells expressing a particular T-cell receptor (TCR)-CD3 complex that recognizes the peptide. Moreover, T helper cell activation causes cytokine production, especially IFN-gamma, to induce B-cell proliferation, differentiation, and Ig production [34,35]. This proves how virosomes induce both humoral and cellular immunity [35]. Compared to the intact virus, the study of the efficacy of Inflexal^®^ V, a licensed influenza vaccine based on a virosomal delivery system, showed good immunogenicity in both healthy and immunocompromised elderly, adults, and children [36]. Figure 3 illustrates the mechanism of action of virosomes.

Bron. et al. used the polypeptide toxin Diphtheria toxin A (DTA) to investigate the ability of virosomes to be encapsulated inside the cells [37]. After internalizing DTA into the hollow virosome, Pyr-PC fluorescent dye was used to mark them. The findings showed that DTA was effectively internalized within the cell’s cytoplasm, indicating that virosomes had a high potential for delivering chemicals to the intended cells.

### 4.2. Stimulation of the Immune System—MHC Activation

Cellular immunity, especially CTL activation, is the main step in clearing viruses from the body. This cannot be applicable except if the antigen of viruses can be encapsulated in the cytosol of APCs, which can then stimulate MHC presentation on the cell surface [38,39]. A study investigated the ability of influenza virosomes to deliver an intact protein antigen, ovalbumin (OVA), for CTL induction in vivo, in which mice received two vaccinations: one with OVA-containing virosomes and one with 100 µg of free OVA. Because of MHC I presentation, OVA-containing virosomes effectively induced OVA-specific CTL activation. However, 100 µg of heat-aggregated free OVA showed poor responses following immunization. Furthermore, the study showed that even 0.25 µg of virosome-encapsulated OVA mounted an adequate CTL response, whereas the smallest antigen dose required to elicit a robust CTL response with free OVA was 0.75 µg [40].

The requirement for fusion activity to effectively induce MHC class II presentation of the encapsulated antigen was examined in a different investigation. OVA-containing fusion-active, fusion-inactive, and dendritic cells (DCs) incubated with OVA in Fc R-targeted or non-targeted liposomes were used as controls for the various DCs. Nevertheless, regardless of fusion activity assessments, the data showed that effective MHC class II presentation of OVA peptide was seen at picomolar doses of OVA in all treated DC [31,41]. According to this, virosomes have an advantage over other vaccines in inducing MHC class I presentation and cellular immune activation due to their fusion activity, which does not affect the induction of MHC II presentation.

## 5. Manufacturing of Virosomes

### 5.1. Preparation of Virosomes

Based on the aforementioned virosome structure, the phospholipid bilayer is regarded as a key constituent. Therefore, acquiring a viral envelope is essential to go forward with the first stage of the manufacturing process. However, viruses belonging to the following families—Flaviviridae, Poxviridae, Retroviridae, Paramyxoviridae, and Orthomyxoviridae—should have an outer coat that is rich in lipids and contains viral spikes. Second, there are exogenous lipids that are being pre-solubilized in solubilizing agents. These lipids can be neutral and charged PL, steroid-derived lipids, or neutral and charged synthetic lipids. Some examples of these include monophosphoryl lipid A (MPLA, GLA), 1,2-dipalmitoyl-sn-glycero-3-phosphocholine (DPPC), 1,2-dioleoyl-3-trimethylammonium-propane (DOTAP), 1,2-dioleoyl-3-dimethylammonium-propane (DODAP), or egg phosphatidylcholine (PC).

Activated or inactivated virus pools are first diluted with phosphate-buffered saline (PBS) and then ultracentrifuged. The resulting pellets are then solubilized using a solubilizing agent, such as a mixture of solubilizing agents, organic solvents like methanol, ethanol, chloroform, dimethylsulfoxide, n-methyl pyrolidinone, tetrahydrofuran, etc., or non-ionic surfactants like TRITON^®^ X-100, Tergitol type NP-40, Octyl glucoside, and Pentaethylene glycol mono-dodecyl ether. The solubilized mixture is then centrifuged once more. However, by adjusting the ratio of spike proteins like NA and HA during these phases, the final virosome’s immunogenicity can be increased by reducing the amount of empty phospholipid envelope.

The solubilized exogenous mixture (PL suspension) and the re-centrifuged mixture were then mixed for 30 to 60 min at room temperature (15 to 25 degrees Celsius) until there were no more visible particles in the mixture [8,42].

The solubilized exogenous mixture (PL suspension) and the re-centrifuged mixture were then mixed for 30 to 60 min at room temperature (15 to 25 degrees Celsius) until there were no more visible particles in the mixture [8,38]. Purification of the resulting virosomes from the solubilizing chemical is required during virosome reconstitution. Batch chromatography (using BIOBEADS^®^), affinity chromatography, gel filtration, dialysis, cross-flow filtering, and other techniques can be used for this. Oligoethylene glycol (OEG) or any other solubilizing agent can be fully eliminated in this way. Following that, a 0.22 µg filter was employed to filter the raw monovalent virosome, which was then kept between 2 °C and 8 °C until it was needed [8,42]. The authorized Inflexal^®^ V trivalent virosome subunit influenza vaccine18 was produced using this manufacturing technique [20]. Figure 4 demonstrates the flow of the manufacturing process.

### 5.2. Preparation of Influenza Virosome

The first vaccination based on a virosomal delivery technology to be authorized was influenza virosomes [8]. One study used muramyl dipeptide derivative (B30-MDP) to construct an influenza virosome vaccine. Because B30-MDP has more chemical stability, stronger immunoadjuvant action, and less toxicity than MDP [43,44]. It was employed as an analog for MDP in the study. Hemagglutinin-neuraminidase antigens (HANA), which create the virosome spikes, were combined in a 1:1 ratio with the liposome’s constituents, cholesterol and B30-MDP, during the manufacturing process. The detergent used to dissolve these components was octyl-beta-D-glucoside, which has a low affinity for proteins and is readily extracted from solutions.

However, the detergent was eliminated using a flow-through dialyzer at room temperature following the reconstitution of the virosome. In mice and guinea pigs, the influenza virosomes showed increased persistence, cellular immunity, and circulating antibody levels [45].

### 5.3. Advances in Manufacturing Methods—Cell-Free Protein Synthesis

Following the mixing of all components, the conventional preparation techniques previously discussed entail a self-assembly mechanism [46]. The addition of detergents in this procedure, however, has the potential to disrupt the virosome’s structural and functional characteristics as well as obstruct its ability to self-assemble and effectively encapsulate medications [47,48]. By avoiding the use of surfactants or detergents, the cell-free protein synthesis (CFPS) method circumvents the drawbacks of conventional manufacturing techniques. CFPS is a simple and fast approach that gives scalable yields that are amenable to modifications [49]. It involves removing the desired cell components and then repeating purification processes. Cell components are collected from a variety of organisms used in CFPS to create the preferred delivery method. These species, which include yeast [50], HeLa, *E. coli*, and rabbit reticulocytes, are referred to as platforms.

The synthesis of virus-like particles (VLPs) was carried out using CFPS to control the disulfide bond formation in VLPs by directly controlling the redox potential during or after production and assembly, using the *E. coli* platform [51]. VLPs are virus envelopes that lack genetic material, which makes them unique for delivery systems, either for drugs or vaccines. A new technique was required to get beyond the restrictions on VLP production, which included low yields and high expenses. The *E. coli* platform was used to create CFPS, and the yield of MS2 bacteriophage coat protein VLP was 14 times higher than that of other techniques [52].

These steps are used in the CFPS method. First, according to the delivery system we wish to create, the proper lysate system is selected. In the meantime, the polymerase chain reaction (PCR) is used to create the base sequence of the desired polypeptide chain and attach it to the expression vector. In order to create cell-free protein in transcription/translation operations, the lysate system is then added to the DNA vector. RNA polymerase is then added along with additional reagents to facilitate DNA transcription to mRNA during cell-free protein synthesis. Lastly, radioactive 14C leucine was added to the process to radiolabel the proteins that were produced [53]. The CFPS batch-based format [49] is the name of this technique [49]. The continuous exchange cell-free synthesis platform (CECF) is another recently developed structure that uses a semi-permeable dialysis membrane to divide the feeding and reaction chambers. The feeding chamber then gradually supplies a reaction chamber with reaction components [49,54,55].

The CFPS approach successfully produced a large number of virosomes, the reticulocyte lysate system created HA2 virosomes, and the resulting virosome demonstrated a noteworthy capacity to carry siRNA [56]. Another example is the creation of virosomes using the MS2 bacteriophage coat protein, which is made from *E. coli* and used in CFPS, as a siRNA carrier [57]. It should be noted that in the CFPS virosome formation process, protein synthesis and membrane assembly take place concurrently, and the vessel [54] is supplemented with the proper lipid ratio for liposome development [47]. Because it creates a final virosome with greater yields, lower costs, shorter time, higher integrity, higher purity, and more control over gene delivery and proteins that can be incorporated on the virosome’s envelope, the CFPS method has advantages over the conventional virosome synthesis method.

## 6. Applications of Virosomes

Virosomes demonstrated a new structure that allows the delivery system to transfer a material to the desired location and trigger an immune response. This makes it an effective delivery method for both vaccines and anti-cancer drugs [8,58,59].

### 6.1. Viral Vaccines

#### 6.1.1. Influenza Vaccines

The influenza virosomal vaccine is a licensed vaccination based on virosomes, while many virosomal vaccines are still in clinical trials [24]. Inflexal^®^ V Berna, an Immunopotentiating reconstituted influenza virosome (IRIV), is the first approved influenza virosome. It has been licensed in Switzerland since 1997 and is currently licensed in 25 countries worldwide. Safety and efficacy findings were obtained from clinical trials involving 2500 healthy individuals. It also showed tolerability in children, and over 10 million vaccine doses have been administered [20]. Inflexal^®^ V owes its activity to the high purity of the influenza glycoproteins intercalated into the phospholipid bilayer, the biological activity of HA, and the natural antigens presented on the virosome surface.

Although the structure of the virosome is innovative and the fusion properties are novel, HA and NA undergo variations whenever a new influenza strain emerges [60,61]. This tackles the need for reformulating new virosomes every year because they cannot protect against new strains. Dong et al. conducted a study in which influenza virosomes were adjuvanted with MPLA, which is a Toll-like receptor 4 ligand, and nickel-chelating lipid, which is a metal-ion-chelating lipid DOGS-NTA-Ni that binds to a histidine-tagged protein on the virosome [22]. Activating cellular immunity is the main approach in virosomal vaccines. Toll-like receptor signals enhance the fusion and phagocytosis [62,63]. As a result, a substance can be encapsulated inside the APCs, and cross-presentation of MHC I occurs. In addition, nickel-containing liposomes play a main role in the association with histidine-tagged protein, an antigen that stimulates antibody response [64,65]. After their method, Dong et al. came to the conclusion that, in comparison to traditional influenza virosomes, virosomes that included MPLA and DOGS demonstrated a considerable activation for APCs on RAW-BlueTM cells. However, virosomes with DOGS alone were used to assess APCs activation, and not enough APCs activation was seen, indicating that DOGS is insufficient on its own.

The 2000-licensed intranasal vaccine Nasalflu^®^ (Berna Biotech, Bern, Switzerland) contains virosomes from three influenza strains: A(H1N1), A(H3N2), and B, along with a heat-labile toxin adjuvant [66]. Despite its safety, humoral immunogenicity, and superior mucosal immunogenicity, Nasalflu was withdrawn after a year of approval because it was estimated that patients who received the vaccination had a 19-fold higher incidence of Bell’s palsy than the controls [66,67].

When compared to the usual influenza vaccination Influvac [17], Invivac^®^ (Solvay, Brussels, Belgium), which contains virosomes from three influenza strains—A(H1N1), A(H3N2), and B—was safe and well tolerated in older adults. It was licensed in 2004. But it has not been available for purchase since 2005 [68]. Given that Nasalflu and Invivac are effective influenza vaccines, this emphasizes the necessity for additional research on their safety before reapproving them.

#### 6.1.2. Hepatitis A Vaccines

Epaxal^®^, the first virosomal vaccine to be authorized in 1994, is made from influenza strain A (H1N1) and coated with the inactivated hepatitis A virus as the target antigen [21]. Unlike the first approved hepatitis A vaccine, which is an adjuvant based on aluminum salts and has a longer half-life in the human body [69], Epaxal^®^ does not cause a nonspecific inflammatory response or local side effects, such as pain and swelling at the injection site in certain people [70]. Additionally, it demonstrated immunogenicity and protection duration, with a median of 55.5 years. Additionally, a 0.25 mL dose of the Epaxal^®^ Junior vaccine for children has been licensed, which has the same safety and tolerability profiles as the adult dose of 0.5 mL [71].

#### 6.1.3. COVID-19

As severe respiratory syndrome coronavirus 2 (SARS-CoV-2) vaccines available have low coverage and availability, the emergence of SARS-CoV-2 variants is a concern [72,73]. Fernandes et al. suggested a virosomal-based vaccine consists of using insect cells to produce a high-yield of spike proteins [74]. The insect cells-baculovirus expression vector system (IC-BEVS) was used to produce high-quality S protein, and this system showed good potential to produce highly glycosylated and complex S protein without affecting the integrity and antigenicity of the protein.

### 6.2. Vaccines in Clinical Development

#### 6.2.1. Hepatitis C

In 2006, Pevion Biotech announced phase I clinical trials for hepatitis C virosome, which were completed in 2008 [75]. The vaccine is composed of a combination of the PeviPRO and PeviTER platforms using synthetic HCV peptide antigens [76]. Although the vaccine showed a humoral immune response, it caused various adverse events. However, to date, the clinical trial has not continued to the next stage. Given the vaccine’s strong initial results, more research might be performed to solve the Hepatitis C virosome’s shortcomings and enhance it in order to see whether it can go to phase II of the clinical trial.

#### 6.2.2. Immunodeficiency Virus Type 1 (HIV)

There is a phase I randomized double-blind controlled study on the HIV virosomal vaccine, MYM-V101. The gp41 protein, a spike found in the HIV envelope, fuses and anchors with the immune cell. It also has an extended MPER region, a binding site to the mucosal receptor galactosyl-ceramide found on dendritic and epithelial cells (P1) [77,78], and the receptor binding domain gp120 [79]. The development of an HIV vaccine benefits from these two proteins. The HIV virosome has given nonhuman primates (NHP) the ability to produce mucosal antibodies against vaginal heterologous virus challenges [78].

After the study, nearly all vaccinees had vaginal and rectal P1-specific IgGs. HIV virosome is a promising vaccine approach for sexually transmitted HIV-1 [79], as evidenced by the detection of vaccine-induced mucosal anti-gp41 antibodies, which is comparable to earlier research on NHP. A barrier to achieving a shelf-life stability of more than two years [74] is the susceptibility of protein and peptide antigens (P1 peptide: virosome-P1 and recombinant gp41: virosome-rgp41) to chemical changes (oxidation and deamination) if stored at a temperature higher than 4 °C [80]. The Manufacturing Process for Cold Chain Independent Virosome-based Vaccines (MACIVIVA) consortium, on the other hand, seeks to create new GMP pilot lines for the production of thermostable vaccines that contain stabilized antigens on influenza virosomes as enveloped virus-like particles [81]. Amacker et al. employed MYM-V202, which is a liquid form of virosomes, as a proof of concept to create a thermostable mucosal solid vaccine form through lyophilization for sublingual tablets [74] or spray drying virosomes for nasal and oral powder [80].

The two different 3M-052 adjuvanted virosomes that make up the liquid HIV-1 candidate vaccine MYM-V202 are the virosomes that contain rgp41 and the virosomes that retain P1 peptides. The manufacturing process for the solid virosome formulation was identical to that of MYM-V202 [78]. Trehalose had to be added in order to maintain the virosome’s integrity while the solid dosage form was being manufactured. Trehalose, a naturally occurring, non-toxic sugar that is frequently employed as a protein stabilizer and cryoprotectant, was a recently chosen excipient added during the in vitro virosome production process, which made this possible [82]. This method was used to develop three pilot lines: spray-dried nasal powder for loading into a dry powder nasal device, lyophilized fast-dissolving sublingual tablets, and spray-dried oral powder for loading into an enteric-coated capsule.

In addition to maintaining the vaccine’s immunogenicity and preserving the majority of the lipid-based virosome structure and antigens with their essential epitopes, they all demonstrated thermostability, supported targeted immune activation, and enhanced vaccine safety and tolerance.

#### 6.2.3. Mumps Vaccine—DNA Virosome

Using the virus DNA [21], Cusi et al. investigated the development of the intranasal Mumps virus (MuV) virosomal vaccination [23]. The Urabe Am9 strain of the mumps virus’s HA and fusion (F) genes were added as components for the spikes that cause virosome fusion in order to create the DNA virosome. In the meantime, the virosome contained the recombinant plasmids GC9 (MuV-HN) and GC23 (MuV-F). Despite not producing neutralizing antibodies against GC23, the vaccinated animals displayed a humoral response and produced neutralizing antibodies against GC9. Furthermore, compared to mice inoculated with live MuV, GC9-virosomes generated a mucosal antibody response with significant levels of IgA; however, GC23-virosomes did not produce any IgA. Lastly, T-helper cell proliferation levels were increased for cell-mediated immunity, and GC9-virosomes triggered a T-helper 2 response because, in addition to IgA, inoculated mice also showed a considerable level of IL-10 and a larger output of IgG1. In contrast, a T-helper 1 response is characterized by the generation of IL-2 and IgG isotyping in mice inoculated with GC23-virosomes. These findings are thought to be encouraging for the creation of innovative DNA virosome-based MuV vaccines. Table 1 summarizes the effects of the two mumps virosomes.

#### 6.2.4. Virosomal Malaria Vaccine

Phase I/IIa trials were conducted for the combined malaria vaccine FFM ME-TRAP and PEV3A. The two malarial antigens that comprise PEV3A, a virosomal vaccine, are derived from the circumsporozoite (CS) protein and apical membrane antigen-1 (AMA-1) of the K1 isolate of P. falciparum. Antibodies produced against CS peptides prevent sporozoite invasion and motility. However, AMA-1 peptides, which resemble loop I and domain III and can generate antibodies, prevent blood-stage P. falciparum parasites from growing [83,84]. Thompson et al.’s results demonstrated that while vaccinated patients demonstrated blood-stage immunity, total protection against malaria did not manifest. The vaccine’s lower rates of parasite development in comparison to controls offered proof of vaccine-induced blood stage efficacy.

### 6.3. Virosomes as a Delivery System of Drugs

The benefits of virosomes are numerous. There are other uses outside vaccination, even though they were created to boost immune responses against certain viruses. Depending on the material’s physical and chemical characteristics, virosomes can be utilized to transport bioactive medications, nucleic acids, genes, or any other protein by either integrating the material on the surface or encasing it inside the hollow virosome [85,86].

#### 6.3.1. Virosomes as a Delivery System for Anti-Cancer

As an anti-cancer treatment, a lot of research is performed on the peptides and antigens that match Tumor-Associated Antigens (TAA), which can be transported by virosomes. Correale et al. examined the responses of particular cytotoxic T lymphocytes (CTLs) to peptides that were carried by influenza virosome and derived from the parathyroid hormone-related protein (PTH-rP) [18]. To create the recombinant plasmid GC90, the PTH-rP was isolated and amplified from the prostate cancer cell line, cloned using expression vectors, and then encapsulated within the influenza virosome. Mice exposed to the virosome containing PTH-rP have cytotoxic efficacy against class I-matched cancer cells that manufacture PTH-rP, as well as a multi-epitopic CTL-mediated immune response.

According to these findings, the tested strategy may be a unique targeted treatment for lung and prostate cancers, as well as those whose epithelial cancers have spread to their bones. Antirat Neu Virosomes, which target the HER-2/neu oncogene, a member of the epidermal growth factor receptor family that encodes a transmembrane growth factor receptor [87], are another use for virosomes as targeted medicine delivery. Metastatic sites also have overexpression of HER-2/neu. Because of this, HER-2/neu is a good target for cancer cell mitigation [88].

As an anti-cancer treatment, Waelti et al. investigated using antirat-neu virosomes to target HER-2/neu [89]. When anti-rNeu-Fab′ fragments were attached to the virosomal envelope and tested on mice, the results demonstrated that, in contrast to unconjugated virosomes, virosomes bound and internalized into tumor cells. Additionally, the experiment was conducted again using a conjugated virosome and encapsulated doxorubicin. The findings indicated that early treatment following tumor inoculation successfully inhibited the formation of tumors in mice.

Despite having a strong, nontoxic, anti-inflammatory, antibacterial, and anti-cancer effect, curcumin derived from Curcuma longa is poorly soluble and bioavailable [90]. To evaluate this delivery system’s capacity to boost bioavailability, lower toxicity, and ultimately boost anti-cancer effectiveness, Kumar et al. created virosomes encapsulating nano-curcumin [91]. The following is a summary of the findings: Nano-curcumin adjuvanted virosomes can improve drug targeting to the preferred site, overcome low solubility, and offer regulated release of medicines.

#### 6.3.2. Anti-Alzheimer Virosomes

The transmembrane protein amyloid precursor protein (APP) is mutated in Alzheimer’s disease (AD), which increases the production of β-amyloid peptides (Aβ peptides), particularly Aβ42, which promotes the formation of amyloid aggregates and plaques, leading to early neurodegeneration and dementia [92]. A unique virosome with an amyloid-peptide-PE conjugate method was demonstrated by Zurbriggen et al. [17]. The immune system was successfully exposed to a molecule based on Aβ1–16 that had a well-defined three-dimensional structure, which resulted in an antibody immunological response with the desired specificity. This virosome could improve cognition and reduce amyloid peptide levels even though it was unable to induce T-cell activation. These results suggest a potential therapy for AD [93,94].

#### 6.3.3. *Candida albicans* Virosomal Vaccine

The most prevalent member of the secretory aspartyl proteinases (Sap) family, Sap 2, is thought to be one of the most important virulence factors causing vaginitis brought on by *Candida albicans* [95,96]. According to the results of an experimental model of *Candida albicans* vaginitis in rats conducted by De Bernardis et al., animals given vaginal fluids containing Sap antibodies were much more protected against vaginitis than rats given vaginal fluids devoid of antibodies [97]. The studies’ findings support the notion of creating vaccines by combining Sap2 subunits. N-terminally truncated Sap2 protein (rtSap2) was added to the virosomal vaccine as an adjuvant by De Bernardis et al. because this protein produced anti-Sap2 IgG and IgA [90] and demonstrated an immunization effect against vaginitis [98]. The resulting virosomes, known as PEV7, were then tested as a potential treatment for chronic recurrent vulvovaginal infection by *Candida albicans* (RVVC) [16]. Nonetheless, the following is a summary of the experiment’s findings: 1. PEV7 provides a consistent level of protection against vaginitis; 2. PEV7 is immunogenic in animal models following intramuscular and intravaginal administration; and 3. PEV7’s safety profiles make the vaccine a strong contender for clinical trials against RVVC.

## 7. Advantages of Virosomes

### 7.1. Industrial Advantages

As stated in the preceding sections, virosomes are a special industrial dosage form for vaccines and medication delivery systems since numerous virosomal applications were explored utilizing a variety of manufacturing techniques. Nonetheless, clinical data for the three hepatitis A virosome injection routes—intramuscular (IM), subcutaneous (SC), and intradermal (ID)—were examined. Additionally, testing of influenza virosomes administered intranasally (IN) revealed that all routes provided high tolerability [99]. As a result, virosomes have an advantage over other vaccines in commercial preparations since they can be administered through a variety of methods.

Another advantage on the industrial level is the comprehensive quality control test, which is recorded in the European Pharmacopeia monographs for the approved products Epaxal and Inflexal V. The first is the quantitative composition analysis, which includes the records’ quantitative analysis of the antigen of interest, influenza virus components, and excipients (such as phospholipids), as well as the lack of dangerous contaminants [68,100]. Particulate architecture, size, and homogeneity are the second factors that may have an effect on vaccine availability in the proximal lymph nodes. They can also be utilized for reliable, repeatable production [20,101]. The third component, virosome functional analysis, examines how virosomes work both in vitro and in vivo, including fusion and immune response. The production of virosomes is facilitated and made more predictable by these recorded quality control checks. This would simplify the production scale-up and speed the necessary in vivo test for new commercial goods [68].

### 7.2. Potency and Efficacy

Vaccination is thought to be the most effective way to stop infectious diseases. However, the circulating infection and the characteristics of the vaccinated individuals (e.g., age, gender, chronic conditions, etc.) have a significant impact on the effectiveness and potency of vaccinations. Furthermore, the type of vaccine (e.g., inactivated, entire virus particle, Split Virus Vaccine, Subunit Vaccine, etc.) affects its potency and effectiveness. For instance, children (ages 6 months to 7 years) responded better to live-attenuated vaccines [94] than did adults [102,103]. This addresses the requirement to develop vaccines that are as effective and potent for all people. Scheifele et al. compared the safety, potency, tolerability, and immunogenicity [96] of the three trivalent inactivated influenza vaccines (Intanza^®^, Agriflu^®^, and Fluad^®^) in a randomized controlled experiment [104,105]. All three experienced post-vaccination injection site pain and redness, but these side effects were well handled; therefore, the results were similar. Additionally, the trial’s tested vaccines had similar baseline antibody titers.

To evaluate the efficacy of Fluad^®^ and Inflexal V^®^ in preventing hospitalization for influenza and pneumonia in the elderly, Gasparini et al. carried out a matched case–control study [106]. Compared to Fluad^®^, the virosomal vaccination (Inflexal V^®^) demonstrated a greater level of efficacy (95.2% and 87.8%, respectively). To examine the safety and immunogenicity of a subunit vaccination (Influvac^®^), virosomal vaccine (Invivac^®^), and adjuvanted vaccine (Fluad^®^) [98], De Bruijn et al. conducted a randomized, endpoint-blind, parallel-group study in aged subjects 61 years and older. None of the three examined vaccines was shown to be less effective or immunogenic [19].

However, Influvac^®^ 11 (8.5%), Invivac^®^ 8 (6.3%), and Fluad^®^ 18 (13.8%) were the patients who experienced at least one treatment-emergent adverse event (TEAE), including arthralgia, sepsis, joint abscess, and psoas abscess. The majority of Influvac^®^ and Invivac^®^ reactions lasted one to two days, but a sizable portion of Fluad^®^ reactions lasted three days. Following vaccination with Invivac^®^ as opposed to Fluad^®^, a statistically significant decrease in the number of local reactions was noted. Virosomal influenza vaccines are more effective and potent than conventional vaccinations, according to the findings of earlier research. Additionally, it was shown that tolerance is higher.

## 8. Patents

### 8.1. Preparation Techniques

We have already discussed manufacturing procedures. On the other hand, virosome preparation is regarded as a unique process that can be used as a drug delivery system for anti-cancer or anti-Alzheimer’s disease treatments, as well as a vaccine delivery system. If the preparation method hadn’t been copyrighted, this wouldn’t have occurred. The process of preparing virosomes is the subject of numerous patents. The European Patent Office (EPO) filed the first patent in 2011 on the creation of virosome particles, which involved generating HA expressed on plants and combining it with phospholipids to create the plant-IRIV [107]. The patent was withdrawn from the EPO in 2013. However, this generation process was submitted in 2013 by the United States and assigned to FRANVAX S.R.L. in 2014 [107]. Another virosome preparation technique was submitted by EPO in 2014 and assigned to Janssen Vaccines and Prevention BV [42]. The process was previously explained in the section on virosome manufacture.

A patent was awarded in 2018 for an enhanced virosome formulation that included ingredients to make the virosomes resistant to temperature in the event of unintentional freezing (2 °C to roughly 8 °C), as this temperature could harm the formulation [108]. The liquid composition of this method is mostly composed of a virosome, trehalose, and a combination of KH_2_PO_4_/Na_2_HPO_4_ buffer with a pH range of 6.5 to 8. This finished product can be stored for more than six months, a year, one and a half years, two years, or longer at 2 °C to 8 °C or ≤−65 °C.

### 8.2. Lyophilization of Virosomes

After being submitted in the US in 2007, the Zurbriggen et al. patents were transferred to HELVETIC AIRWAYS AG in 2015 [109]. Putting virosomes in a vacuum can change them from a solid phase into a gaseous phase (vapor) without causing them to transition into an aqueous phase. This process, known as lyophilization, is the process of freezing and then sublimating a product to remove water [110]. The lyophilization process is used to change the virosomes’ composition in order to produce a better, more resilient freeze-drying virosome. In this case, the virosomes are made up of biologically active substances (such as immunogenic or pharmacological substances) and cationic lipids (cationic cholesteryl derivatives), which are found in the virosome membrane and aid in the efficient lyophilization and reconstitution of the virosome. This innovation offers the benefit of adding adjuvants such as Freund’s (full and incomplete), mycobacteria like BCG, M. Vaccae, or Lipid A, or Corynebacterium parvum, which can significantly increase immunogenicity and potentiate immune response. Additional adjuvants include mineral salts or mineral gels, LPS derivatives, saponins, and surface-active substances such as peptides, protein fragments, pluronic polyols, lysolecithin, and polyanions.

Numerous preparations, including immunopotentiating reconstituted influenza virosomes (IRIV), DOXORUBICINE-DIRIV, liposomes containing DC-Chol (DC liposomes), and numerous other preparations, were used to test the lyophilization of the virosome patent [109].

### 8.3. Virosomal Vaccines

Many vaccines that use virosomes as the delivery method have also received patents. A vaccination was required to prevent the spread of the infectious virus known as alphavirus, which is an enveloped single-stranded positive-sense RNA virus that is found all over the world [111]. The RNA replicon systems, which contain nucleic acid sequences encoding antigens for generating an immune response to HIV [105], are the primary components of the virosomal alphavirus vaccine, which was filed in the United States in 2004 [112]. Furthermore, another HIV vaccine patent was submitted in 2007 and awarded to Pevion Biotech Ltd. [106]. This invention is primarily made up of gp41-derived antigen or an analog of the antigen that is present on the outside of or enclosed in the virosomal vesicle, and it has been shown to effectively elicit an immune response against HIV [113].

The Hepatitis A vaccine was prepared using reconstituted influenza virosomes; Cilag GmbH International [107] is presently the owner of the 1993 patent [114]. In order to clarify immunogenicity potentiation, the vaccine’s Hepatitis A virus (HAV) antigen was adsorbed on the virosome surface. Additionally, the Hepatitis B vaccine was created using influenza virosomes; Berna Biotech AG filed the patent application in 2004, and Cilag GmbH International [108] is presently the owner of the invention. The influenza virosome’s liposomal surface, which connects the nucleocapsid protein (HBc) to the virosome structure, was modified in this invention to add the Hepatitis B envelope protein (HBsAG) [115]. Finally, in 2005, a virosome patent was submitted, which had influenza and respiratory syncytial virus (RSV) antigen adjuvanted with saponin to provide high immunogenicity [116].

### 8.4. Delivery of Genetic Material

In 2001, the United States granted a patent for the implementation of the cationic virosome as a genetic material delivery system [117,118]. A positively charged virosome, which is the invention’s main component, is capable of efficiently delivering materials to cells. Viral fusion protein, cell-specific biomarkers like fragments F(ab′)2 and Fab′, cytokines, and growth factors, as well as cationic lipids like N-(1,2,3-dioleoyloxy)propyl-N,N,N-trimethylammonium chloride (DOTMA) and N-(1,2,3-dioleoyloxy)-propyl-N,N,N-trimethylammonium methylsulfate (DOTAP), and cytokines allow the virosome to bind to the target cell selectively. This innovative technique makes virosomes a great approach to delivering active pharmaceutical ingredients for cancer and leukemia treatments.

For further information, cationic virosomes were manufactured that contained genetic material such as pcDNA3 with the human IL-6 gene and antisense L-myc-FITC oligodeoxynucleotides. The target cells were more readily attached and penetrated by these virosomes. This happened as a result of interactions between nucleic acids and the positively charged lipid bilayer, which led to their concentration inside the vesicles. The findings demonstrated that encapsulated genetic material, such as antisense-L-myc, was 20,000-fold more active than antisense L-myc that was not enclosed. The absence of safety and stability investigations on cationic virosomes, however, emphasizes the necessity of more investigation before they can be used as a possible treatment. The patents discussed are summarized in Table 2.

## 9. Future Prospects

As conventional anti-cancer drugs possess high susceptibility to resistance upon long-term use and low selectivity, which causes side effects such as cardiotoxicity and alopecia [120,121]. The need for emerging and novel anti-cancer therapies has risen.

A novel anti-cancer therapy is under clinical trials, which involves anti-cancer peptides (ACPs) that have shown good activity on breast cancer, hepatocellular carcinoma, and cervical cancer [122]. APCs are bioactive peptides derived from natural animal and plant sources, and they exhibit antimicrobial, antioxidant, and anti-cancer activities. As they are composed of a large proportion of both hydrophobic and positively charged residues, they interact with the negatively charged tumor cell membrane, causing cytotoxic activity [123,124].

However, these APCs have low stability, high degradation susceptibility, and low bioavailability [122]. Low permeability is one of the limitations faced in ACPs. This limitation can be solved if cell penetration peptides (CPPs), which are hydrophobic in nature, are mainly composed of basic residues and play an important role in the interaction and insertion of peptides into the cell membrane [125], are conjugated at the surface of virosomes, which increases the ability of the ACPs to fuse with the cancer cells and, as a result, increases permeability. Moreover, it would be promising to avoid the poor bioavailability and stability of ACPs dosage forms by combining the ability of virosomes to transport material within their hollow cavity or onto their envelope to target tumor cells with novel techniques for the preparation of thermostable solid dosage forms [80]. The thorough efforts presented in this study suggest that additional research on anti-cancer virosomes is necessary in order to develop new anti-cancer therapeutics with high bioavailability.

As previously mentioned, the new, innovative anti-cancer treatment employing ACPs may benefit from the possible application of cationic virosomes to transport genetic materials. We can utilize machine learning to forecast the safety and stability of cationic virosomes, notwithstanding their shortcomings. Furthermore, machine learning would provide researchers with valuable information on how to leverage the effectiveness of cationic virosomes and perhaps use them as a vehicle for delivering the innovative ACP method.

## 10. Conclusions

The review concludes that virosomes are a unique drug delivery technology that may find application outside vaccines in a variety of fields. Promising trials on many vaccines, cancer therapies, and other oral medications are conducted to check the potential use of virosomes as a delivery system.

In addition to patents awarded for virosome-based anti-cancer therapies, other studies showed promise for employing virosomes as a cancer treatment delivery mechanism. The bioavailability of oral anti-cancer medicines is low and varies, primarily due to cytochrome P450 (CYP) activity and drug transporters, such as P-gp [126]. Higher therapeutic results and enhanced bioavailability are possible if these circumstances can be avoided.

The flexibility of the virosomal delivery system components makes it a novel delivery system for many medications. Moreover, the safety profile of patented virosomes as vaccines or as a delivery system for genetic material encourages researchers to study their potential use for treatments that have a history of high side effects. This review highlights the importance of studying the possible use of virosomes as a solution for the treatment of diseases that are still untreated. Especially while working on the review article, it was found that there is a lack of recent studies, opening the chance to work on new research to enhance safety, immunogenicity, and stability of the virosomes.

## Figures and Tables

**Figure 1 life-15-01567-f001:**
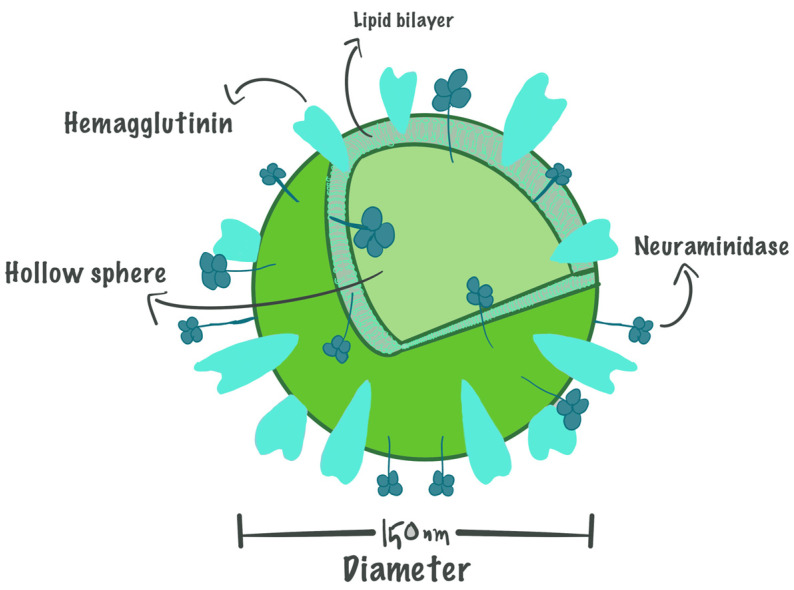
The basic structure of virosomes based on the structure of the Influenza envelope.

**Figure 2 life-15-01567-f002:**
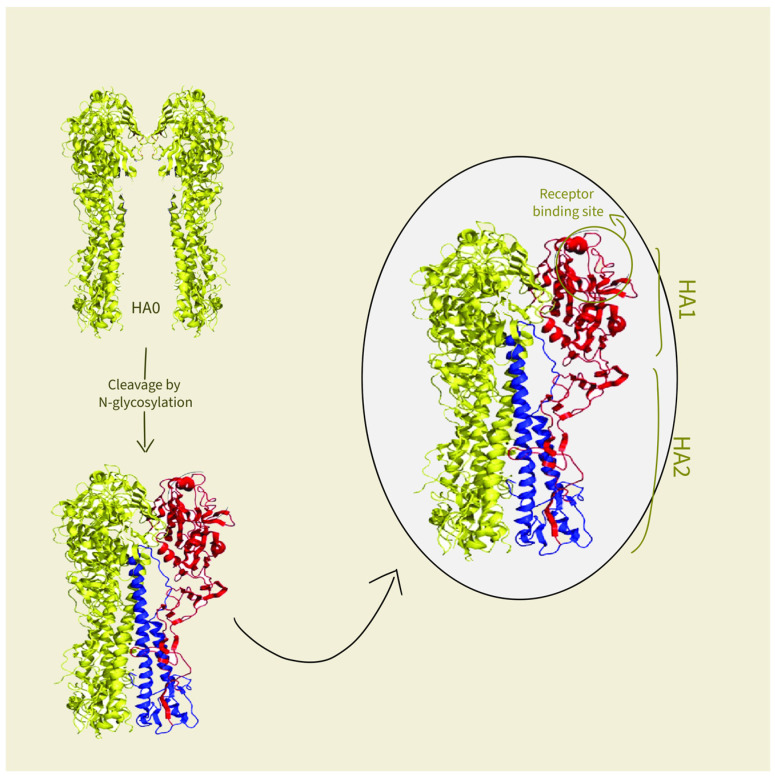
Hemagglutinin glycosylation and cleavage.

**Figure 3 life-15-01567-f003:**
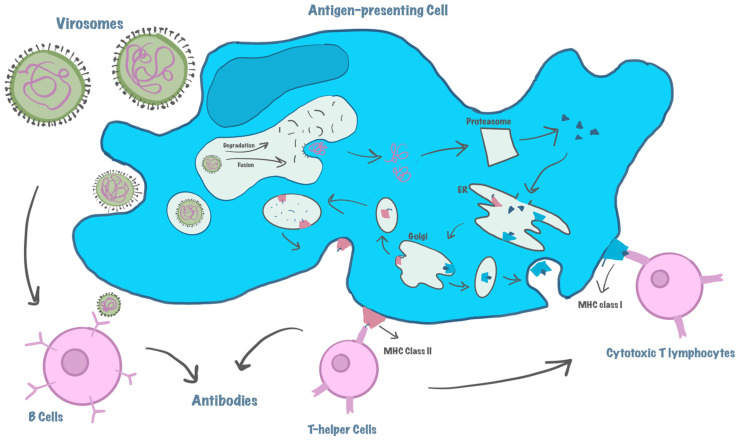
An example demonstrates how a virosome engages in interactions with cells that present antigens and triggers an immune response.

**Figure 4 life-15-01567-f004:**
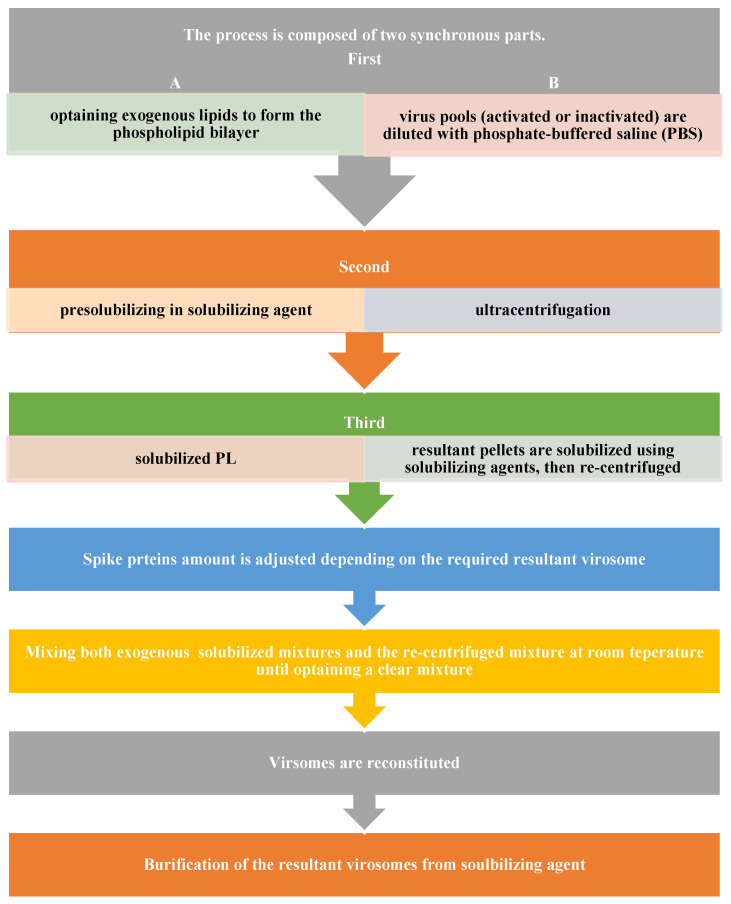
Conventional virosomal manufacturing process.

**Table 1 life-15-01567-t001:** The distinction between the GC9 and GC23 virosomes’ immunological responses.

Mumps Virosomes	Mucosal Immunity Response	Cellular Immunity Response
GC9-Virosome	Instigated a humoral response with a significant increase in IgA production.	Triggered T-helper cell 2 response, characterized by increased levels of IL-10 and IgG1
GC23-Virosome	No response	Triggered T-helper cell 1 response, characterized by increased levels of IL-2 and IgG

**Table 2 life-15-01567-t002:** Summary of useful patents on virosomal delivery systems.

Virosome Patent	Novel Composition	Current Assignee	Patent Office	Legal Status
Patent on the preparation Technique				
Generation of a virosome particle	Deriving HA expressed on plants and mixing them with phospholipids to generate the plant-IRIV89.	FRANVAX Srl	United States	Abandoned
Method for preparing virosome	Virosome reconstitution using a solubilizing agent, followed by purification by BIOBEADS	Janssen Vaccines and Prevention BV	European Patent Office (EPO), and the United States	Active
Improved formulation of virosomes	Adding KH_2_PO_4_/Na_2_HPO_4_ buffer at a pH ranging between 6.5 and 8, and trehalose during manufacturing, so that the final product could withstand freezing	Janssen Vaccines and Prevention BV	EPO	Active
Lyophilization	Lyophilization gives the advantage of adding adjuvants that can enhance the immune response	HELVETIC AIRWAYS AG	United States	Abandoned
Virosomal vaccines				
Alphavirus	RNA replicon systems, which contain nucleic acid sequences encoding antigens for eliciting an immune response to HIV	University of North Carolina at Chapel Hill, Alphavax Inc.	Australia [119], then the United States	Abandoned
HIV	Virosome-like vesicles comprising gp41-derived antigens	Pevion Biotech Ltd., Institut National de la Sante et de la Recherche Medicale INSERM, Mymetics Corp.	United States	Active
Hepatitis A	Hepatitis A virus (HAV) antigen was adsorbed on the surface of the virosomes so it could elucidate immunogenicity potentiation	Cilag GmbH International	United States	Expired
Hepatitis B	HBsAG was incorporated in the liposomal surface of the influenza virosome, which is used to link the HBc to the virosome structure	Cilag GmbH International	EPO	Withdrawn
Influenza and RSV	Vaccine compositions comprising virosomes and a saponin adjuvant	GlaxoSmithKline Biologicals SA	EPO	Withdrawn
Delivery of genetic materials				
Anti-cancer, leukemia	Cationic virosomes as a transfer system for active pharmaceutical ingredients	Nika Health Products Ltd.	United States	Expired

## Data Availability

The original contributions presented in this study are included in the article. Further inquiries can be directed to the corresponding author.

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
