# Peer review of "Virosomes: Beyond Vaccines"

_life, 2025, doi:10.3390/life15101567_

Round 1
Reviewer 1 Report
Comments and Suggestions for Authors
Authors need to resubmit the article after addressing the following concerns;
- Very poor resolution of figures
- Connectivity of the content is missing
- Please change the title of the manuscript
- Vaccine and drug delivery are discussed in the same section
- Improve the novelty
- Include the recent studies
Comments on the Quality of English Language
NA
Author Response
|
|
Response to Reviewer 1 Comments
|
||
|
|
1. Summary |
|
|
|
|
Your critique of this work is greatly appreciated. Please refer to the resubmitted paper and the thorough responses below.
|
||
|
|
2. Point-by-point response to Comments and Suggestions for Authors All the responses are highlighted in yellow in the revised version.
|
||
|
|
Comments 1: Very poor resolution of figures |
||
|
|
Response 1: I appreciate you bringing this up. We concur with this statement. As a result, we have changed all of the manuscript's figures to 4K resolution. Kindly review them in the manuscript. |
||
|
|
Comments 2: Connectivity of the content is missing |
||
|
|
Response 2: I appreciate your comment. We made a few changes to improve the content's connection. Nonetheless, the sections are organized in ascending order, beginning with an explanation of the structure and mode of action of virosomes, moving on to applications and patents, and finishing with prospects. |
||
|
|
Comments 3: Please change the title of the manuscript |
||
|
|
Response 3: I appreciate you bringing this up. The title of the manuscript has been modified to "Virosomes: Beyond Vaccines." Since the review paper emphasizes that virosomes were used as a delivery strategy other than vaccination, this title seems more appropriate for the manuscript. |
||
|
|
Comments 4: Vaccine and drug delivery are discussed in the same section |
||
|
|
Response 4: Thank you for your thoughtful comments. According to what we understand, we included a drug administration technique in the same section as several vaccinations. The anti-malaria vaccine paragraph was therefore shifted from the part on "virosomes as a drug delivery for drugs" to the section on "virosomes in clinical development." However, the immunizations and drug delivery cannot be discussed in the same section since we want to stress that virosomes are used to treat a variety of diseases other than viral infections, such as cancer and Candida albicans. |
||
|
|
Comments 5: Improve novelty |
||
|
|
Response 5: We made every effort to enhance the novelty. Thus, "see figure 3" was added as a new figure, and "see table 1" was inserted as a new table for further explanation. The prospects’ part was also enlarged, and the gaps in the research on virosomes, including safety and efficacy, were emphasized. Lines 409–411 and 382–354 are two examples.
|
||
|
Comments 5: Included recent studies |
|||
|
Response 5: Although we nearly included every recent study that is pertinent to the subject, you may notice that there is significantly less recent research in the references than older ones. 3. Response to Comments on the Quality of the English Language An American native checked the syntax and flow of the English language. |
|||
Reviewer 2 Report
Comments and Suggestions for Authors
- It is recommended to present this text as a figure. “In the beginning, HA is synthesized as a precursor HA0, and the host cell is responsible for its cleavage into HA1 and HA2 subunits by N-glycosylation. The amino-terminal of the HA1 subunit contains the receptor-binding site on its globular head, bonded to the carboxy-terminal fragment HA2, which is membrane-anchored and responsible for fusion".
- The reference numbers should be placed in brackets, e.g., [8], [31].
- Line 175 and 178: The statement “for CTL induction in vivo, where mice were immunized twice, once with 100g of free OVA” should be revised to correct the quantity formatting to “100 µg” (micrograms) instead of “100g”.
- It is recommended to add a Quality control tests section for virosome products when introducing new commercial products.
- It is recommended to include a Summary of commercial products based on the virosomal delivery system, with explicit mention of their structural properties.
Author Response
|
Response to Reviewer 2 Comments
|
||
|
1. Summary |
|
|
|
Your critique of this work is greatly appreciated. Please review the revised manuscript and the thorough replies and related changes below. |
||
|
2. Point-by-point response to Comments and Suggestions for Authors All responses are highlighted in Red in the revised version.
|
||
|
Comments 1: It is recommended to present this text as a figure. “In the beginning, HA is synthesized as a precursor HA0, and the host cell is responsible for its cleavage into HA1 and HA2 subunits by N-glycosylation. The amino-terminal of the HA1 subunit contains the receptor-binding site on its globular head, bonded to the carboxy-terminal fragment HA2, which is membrane-anchored and responsible for fusion". |
||
|
Response 1: We appreciate you bringing this up. We concur with this statement. As a result, we created a new figure, which is shown in the manuscript as figure 3. |
||
|
Comments 2 and 3: - The reference numbers should be placed in brackets, e.g., [8], [31]. - Line 175 and 178: The statement “for CTL induction in vivo, where mice were immunized twice, once with 100g of free OVA” should be revised to correct the quantity formatting to “100 µg” (micrograms) instead of “100g”. |
||
|
Response 2 and 3: Done
|
||
|
Comments 4: It is recommended to add a Quality control tests section for virosome products when introducing new commercial products. |
||
|
Response 4: We appreciate your suggestion. Nevertheless, we attempted to include the quality control tests, but no such data was published. Comments 5: It is recommended to include a Summary of commercial products based on the virosomal delivery system, with explicit mention of their structural properties. Response 5: We appreciate your input. A table outlining the patents and their characteristics is already included in the article. The section has already covered commercial products. |
||
Reviewer 3 Report
Comments and Suggestions for Authors
Article is suitable for publication after addressing the following comments.
The article would benefit from a thorough grammar checking
1-The current title emphasizes virosomal drug delivery, but the manuscript covers broader applications, including vaccines and cancer therapies. The title should be revised to reflect this broader scope.
2-"Targeting proteins that are difficult to mutate, such as liposomes, nanogels, microemulsions, etc., recent innovative techniques for vaccine delivery systems have surfaced that exhibit excellent immunogenicity features" those are delivery systems not proteins. Please rephrase the entire sentence for scientific accuracy and clarity.
3-In the last lines of the introduction (around Lines 83–83), please expand on what differentiates this review from previous reviews on virosomes.
4-Correct all typos in Figure 3, and remove extra space from line 298
5-Line 318-325 The discussion of Nasalflu’s side effects is redundant. Please merge and streamline these sentences
6-Section 6.2.1 Expand on the Phase I clinical trials: did the vaccine proceed to later stages? What challenges specific to HCV limit vaccine development?
7-Section 6.2.2 Lines 374-381, Paraphrase this section for clarity. Place stronger emphasis on the differences between the liquid and solid formulations and highlight the role of trehalose in stabilizing the solid form.
8-Section 6.2.3 Could better be presented in a table
9-Line 436, use brackets for the reference number
10-Section 6.3.3 Clarify the benefit of the absence of T-cell reactivity in this case
11-Section 7.1 Rewrite this section with a clearer and simpler structure. Explicitly state the three areas of quality control: (1) quantitative composition analysis, (2) particulate architecture, and (3) functional analysis. Ensure logical flow.
12-Correct the subsection numbering: this should be Section 7.2 (and review all subsections for consistency).
13- Section 8.2 "The act of removing water from a product by putting it in a vacuum can transform it from a solid phase into a gaseous phase" The description of lyophilization is incomplete. Emphasize that the process begins with freezing, followed by sublimation. This will help readers understand where the “solid phase” originates.
14-Section 8.4 Expand on the mechanism: what components impart the positive charge (which cationic lipids)? Clarify the role of virosomes in delivering genetic cargo such as DNA, mRNA, siRNA with examples, rather than drugs. Add a comparison with viral vectors, highlighting differences in manufacturing, safety, immunogenicity, and stability.
15-Future Prospects: Broaden this section to discuss additional aspects such as Regulatory challenges for virosome-based therapeutics and the potential role of AI/ML in optimizing antigen design, delivery systems, and manufacturing.
Author Response
|
Response to Reviewer 3 Comments
|
||
|
1. Summary |
|
|
|
Your critique of this work is greatly appreciated. Please refer to the thorough answers, related edits, and highlighted changes in the updated version.
|
||
|
2. Point-by-point response to Comments and Suggestions for Authors All responses are highlighted in green in the revised manuscript
|
||
|
Comment 1: The current title emphasizes virosomal drug delivery, but the manuscript covers broader applications, including vaccines and cancer therapies. The title should be revised to reflect this broader scope.
|
||
|
Response 1: Thank you for pointing this out. We have changed the title to “Virosomes: beyond vaccines”.
|
||
|
Comments 2-5, 9, and 11-15: 2-"Targeting proteins that are difficult to mutate, such as liposomes, nanogels, microemulsions, etc., recent innovative techniques for vaccine delivery systems have surfaced that exhibit excellent immunogenicity features" those are delivery systems not proteins. Please rephrase the entire sentence for scientific accuracy and clarity.
3-In the last lines of the introduction (around Lines 83–83), please expand on what differentiates this review from previous reviews on virosomes.
4-Correct all typos in Figure 3, and remove extra space from line 298
5-Line 318-325 The discussion of Nasalflu’s side effects is redundant. Please merge and streamline these sentences 9-Line 436, use brackets for the reference number 11-Section 7.1 Rewrite this section with a clearer and simpler structure. Explicitly state the three areas of quality control: (1) quantitative composition analysis, (2) particulate architecture, and (3) functional analysis. Ensure logical flow. 12- Correct the subsection numbering: this should be Section 7.2 (and review all subsections for consistency). 13- Section 8.2 "The act of removing water from a product by putting it in a vacuum can transform it from a solid phase into a gaseous phase" The description of lyophilization is incomplete. Emphasize that the process begins with freezing, followed by sublimation. This will help readers understand where the “solid phase” originates. 14-Section 8.4 Expand on the mechanism: what components impart the positive charge (which cationic lipids)? Clarify the role of virosomes in delivering genetic cargo such as DNA, mRNA, siRNA with examples, rather than drugs. Add a comparison with viral vectors, highlighting differences in manufacturing, safety, immunogenicity, and stability.
15-Future Prospects: Broaden this section to discuss additional aspects such as Regulatory challenges for virosome-based therapeutics and the potential role of AI/ML in optimizing antigen design, delivery systems, and manufacturing. |
||
|
Responses to comments 2-5, 9, and 11-15: I appreciate your insightful remarks. We made the necessary changes. Kindly review the revised manuscript file. Comments 6: Section 6.2.1 Expand on the Phase I clinical trials: did the vaccine proceed to later stages? What challenges specific to HCV limit vaccine development? Response 6: We appreciate your input. We made the necessary adjustments. As stated below: "The vaccination caused a number of negative outcomes even though it demonstrated a humoral immune response. The clinical trial hasn't advanced to the next phase yet, though. Given the vaccine's promising early results, more research might be done to address the Hepatitis C virosome's shortcomings and enhance it in order to see whether it can advance to phase II of the clinical trial. |
||
|
Comment 7: Section 6.2.2 Lines 374-381, Paraphrase this section for clarity. Place stronger emphasis on the differences between the liquid and solid formulations and highlight the role of trehalose in stabilizing the solid form Response 7: We totally Agree. The text was changed as directed. See the changes in the revised manuscript |
||
|
Comment 8: Section 6.2.3 Could better be presented in a table Response 8: Thank you for your response. Please see Table 1 in the manuscript. Comment 10: Section 6.3.3 Clarify the benefit of the absence of T-cell reactivity in this case Response 10: Thank you for your feedback. It was changed as follows: “Although this virosome failed to instigate T-cell reactivity, it still could enhance cognition and lower amyloid peptide levels. These findings point to a possible AD treatment”. It is important to note that there was no T-cell response; this is not advantageous. For clarity, it was rewritten. 3. Response to Comments on the Quality of the English Language A native American examined the grammar and flow of the English language. |
||
|
|
||
|
|
||
|
|
||
Round 2
Reviewer 1 Report
Comments and Suggestions for Authors
Thanks for the revision